# The Importance of Foot Function Assessment Using the Foot Function Index-Revised Short Form (FFI-RS) Questionnaire in the Comprehensive Treatment of Patients with Rheumatoid Arthritis

**DOI:** 10.3390/jcm11092298

**Published:** 2022-04-20

**Authors:** Radosław Rutkowski, Małgorzata Gizińska, Małgorzata Gałczyńska-Rusin, Magdalena Paulina Kasprzak, Elly Budiman-Mak

**Affiliations:** 1Department of Physical Therapy and Sports Recovery, Poznan University of Physical Education, Królowej Jadwigi Str. 27/39, 61-871 Poznan, Poland; gizinska@awf.poznan.pl; 2Department of Orthodontics and Temporomandibular Disorders, University of Medical Sciences, Bukowska 70, 60-812 Poznan, Poland; m.galczynskarusin@gmail.com; 3Chair and Department of Medical Chemistry and Laboratory Medicine, Poznan University of Medical Sciences, Rokietnicka 8, 60-806 Poznań, Poland; magdarut@ump.edu.pl; 4Center of Innovation for Complex Chronic Healthcare (CINCCH), Hines VA Hospital, 5000 South 5th Ave, Hines, IL 60141-3030, USA; elly.mak@va.gov; 5Department of Medicine, Stritch School of Medicine, Loyola University of Chicago, Maywood, IL 60513, USA

**Keywords:** FFI, ultrasonography, foot function, rheumatoid arthritis, foot pain

## Abstract

Background. Foot problems may have a substantial negative impact on rheumatoid arthritis (RA) patients’ mobility. They affect walking and the functional capacity to perform daily tasks. Methods. This study included 61 patients with RA and foot pain or swelling. The study group comprised 37 patients (aged 54.3 ± 9.5 years) with foot lesions, as demonstrated in an ultrasound, and the control group comprised 24 patients (aged 57.3 ± 11.5 years) without foot lesions. The patients’ health statuses were evaluated with the Foot Function Index-Revised Short Form (FFI-RS), the Polish version of the Health Assessment Questionnaire-Disability Index (HAQ-DI), and the Disease Activity Score 28 (DAS 28). Results. The FFI-RS showed significant differences between the study and control groups in total results, as well as in the pain and stiffness subscales. Subsequent analyses showed numerous significant correlations. The FFI-RS total results correlated with the HAQ’s standing up, walking, and total results. The FFI-RS pain results correlated with the social issues and HAQ’s total results. The FFI-RS difficulty results correlated with the disease’s duration. In the study group, there were significant correlations of the FFI-RS stiffness, difficulty, and social issues results with the HAQ’s standing up, walking, and total results, and also of the FFI-RS activity limitation results with the HAQ’s standing up results. In the control group, there were correlations of the FFI-RS stiffness, difficulty, and activity limitation results with the HAQ’s walking and total results. Finally, in the study group, we also found correlations of the FFI-RS total, pain, stiffness, difficulty, and social issues results with the Visual Analog Scale (VAS) results, as well as of the FFI-RS total results with the DAS 28 results. Conclusions. The FFI-RS is an effective tool for assessing RA patients’ functional status and can be used to evaluate treatment effects. The FFI-RS detected RA-related changes in the foot joint function in patients without foot lesions, as assessed by ultrasound.

## 1. Introduction

Rheumatoid arthritis (RA) is a chronic autoimmune condition resulting in symmetrical arthritis, sometimes with additional systemic signs or symptoms [1]. Most often, the first clinical presentation of RA is symmetrical pain, swelling and stiffness of joints, often the feet joints [2]. The disease’s frequency is two to four times greater in women than in men, and it is the least common in young men [3].

The feet are one of the most common sites of pathology in RA [4]. When first diagnosed, 53% of people with RA may have foot symptoms [5], which increases to up to 90–100% along with the disease’s duration [6]. The foot problems develop rapidly, with half of RA patients having foot involvement within the three years since the diagnosis. This may have a substantial negative impact on RA patients’ mobility, such as difficulty with walking [7] and lowered functional capacity to perform daily tasks [8]. The function impairment of multiple joints negatively affects RA patients’ quality of life [9].

Identifying foot signs and symptoms related to RA is crucial for early onset of treatment to prevent deterioration. Foot pain and joint stiffness are common symptoms and should always be assessed carefully to detect possible background early [10]. Both the structural and functional derangement often result in impaired capacity to perform physical activities [11]. Although RA patients may have periods of remission, many still suffer from disability and reduced participation in daily living activities [12]. RA-related disability and reduced physical fitness can result in an increased fear of falling, thus reducing the quality of life and activity participation, thus increasing that disability [6].

Our observations show that the changes visible in ultrasound do not reflect the functional state of the feet, which affects the general condition of patients. Therefore, to accurately evaluate the functional changes in the feet, which are known to affect the quality of life in patients with RA, we should use scientifically recognized tools to make the diagnosis process complete. One of these tools is the Foot Function Index-Revised Short Form questionnaire (FFI-RS), a short version containing 34 questions. This version includes five subscales for pain, stiffness, difficulty, activity limitation, and social aspects. According to the authors, the short version has sound psychometric measures [13,14,15]. This study aims to prove usefulness of the FFI-RS Polish Version questionnaire with all of it subcategories in assessing RA patients’ functional status of feet with various clinical changes in feet ultrasound imaging and their correlation with the general condition of the patients.

## 2. Materials & Methods

### 2.1. Participants

This study included 61 patients admitted to a Rheumatology Department, with RA diagnosed according to ACR/EULAR (American College of Rheumatology/European League Against Rheumatism) criteria dated 2010 [16]. In the basic characteristics for each of them, we have defined, among others, the following data: age, gender, disease’s duration, comorbidities, DAS 28, HAQ-DI, and VAS. Additionally, we performed feet-surface skin temperature analyses. The next step of the research procedure was the examination which used the FFI-RS questionnaire. The obtained results were statistically analyzed.

The inclusion criteria were pain and/or swelling of the foot joints. The exclusion criteria were cognitive, proprioceptive or sensory impairment, and a recent foot injury and/or foot bone fracture.

All the participants underwent an ultrasound examination of the foot joints for signs of inflammation, and they were divided into two groups: the study group, which consisted of 37 patients with lesions, as demonstrated on the foot ultrasound, and the control group, which consisted of 24 patients with normal foot ultrasounds.

Each participant was familiarized with all the procedures, and they provided written informed consent prior to inclusion to the study.

The study protocol was approved by the Bioethical Committee at the Poznan University of Medical Sciences, under protocol number 183/14.

### 2.2. Data Collection

Each of the subjects underwent the following research procedures:

Disease Activity Score 28 (DAS28) included the number of swollen and tender joints, the global VAS score assessed by the patient (general health), and erythrocyte sedimentation rate (ESR). Response criteria have been extensively validated and are finding wide range of applications in RA clinical trials and to monitor individual RA patients. The available thresholds define absolute DAS 28 scores representing remission (<2.6), mild (≤3.2), moderate (>3.2), and severe (>5.1) activity of the disease [17]. The DAS28 was used, even though it does not take foot joints into account, because this scale was used at the department to assess the disease’s activity.

Health Assessment Questionnaire-Disability Index (HAQ-DI). To examine physical function, the HAQ-DI was used. The HAQ-DI is a predictive factor of future disability and joint damage in patients with RA. Because it demonstrated sensitivity to change, the HAQ-DI was recommended by the American College of Rheumatology (ACR) to be incorporated into the core set of outcome measures of RA disease activity. The HAQ-DI not only is considered an essential measure of disability in patients with RA in clinical trials, but also is used in clinical practice. It comprises 20 detailed questions about daily activities, divided into 8 categories: dressing and taking care of appearance, standing up, eating, walking, hygiene, reaching, gripping, and daily life activities [18].

The Visual Analogue Pain Scale. The visual analog scale (VAS) is a valid and reliable measure of chronic pain intensity. This is a simple and commonly used tool to be used by anyone cognitively capable of understanding the parameters and responding to a clinician’s instructions. The VAS pain scale results were obtained by measuring the distance in millimeters from the beginning of the scale to the position selected by the patient from “0” to “100 mm”, where “0” is for “no pain” and “100” is for “the worst possible pain” [19].

Foot Function Index-Revised Short Form Polish Version (FFI-RS). The FFI-RS Polish version contains 34 questions in five subscales: pain, stiffness, difficulty, activity limitation, and social aspects. Each participant was coached to correctly fill the questionnaire. Scores range from 0–100, with higher scores representing a worse foot function. This questionnaire is a reliable and frequently used tool in patients with osteoarthritis and rheumatological diseases. In 2017 we adapted and validated the Foot Function Index-Revised Short Form into Polish. The person reliability of the Polish version was 0.95 [20].

Ultrasound. Ultrasound is a quick and inexpensive way to detect synovitis, tenosynovitis, tendon tears, and bursitis. However, it is operator-dependent and also depends on the quality of the USG machine. The ultrasound examinations were performed using MyLab Twice 6–18 MHz transducer, Esaote. The ultrasound was performed by a rheumatologist with expertise in ultrasound. It was performed on the dorsal side of the foot and involved the 1–5 MTP joints. During the test, the foot was in a neutral position. The rheumatologist examined the joints for the presence or absence of signs of inflammation in the form of synovitis and swelling. The presence of inflammation in a joint was defined as a minimal hypertrophy of the synovium and/or slightly increased vascularization [21,22].

Thermovision. Thermovision techniques have been used for various purposes in medicine, including clinical testing of drugs, the assessment of vascular reactions in hands, the diagnosis of Raynaud’s syndrome, the evaluation of observational changes in osteoarthritis, the detection of different kinds of tumors, and the assessment of the skin condition in diabetic feet. We used thermovision to analyze the surface skin temperature of feet–the region of interest (ROI). On the day of the thermal imaging, the patients were not allowed to smoke, drink alcohol or coffee, or use other stimulants and drugs, except for drugs prescribed by their attending physician, which were used in a consistent dosage. Nor were the participants allowed to undertake intense physical activity or use physical treatment prior to the examination. Each participant adapted to the examination room’s conditions for 20 min by uncovering the treated area. The thermal images were captured with the ThermaCAM SC640 (Flir) according to the guidelines provided by the European Association of Thermology [23]. The emissivity was defined at 0.98. The camera was calibrated for 20 min before the first analysis. The imaging was performed in the morning, at the same time. The temperature in the room was maintained 21 °C and humidity 40 ±  10%. The camera was positioned on a tripod 50 cm above the feet, and it was perpendicular to them. The feet were in a natural position. The images of the dorsal side of the feet were captured and used for analysis. The ROI was limited by the outlines of a foot. The proximal end of the ROI was a horizontal line aligned at the tip of the navicular bone.

Statistical analysis. The statistical calculations were conducted using the SPSS v14 software (Statistical Product and Service Solutions). Descriptive statistics were used for determining mean values, standard deviations (SD), and the minimum and maximum of the demographic variables. The normality of the data distribution was checked with the Shapiro–Wilk test. The Pearson’s and Spearman’s correlation tests (depending on the normality of the data distribution) were used to determine correlations between the FFI-RS scores, HAQ-DI, DAS28, ESR, VAS pain, and surface skin temperature of the right/left foot. To test the differences between the study group and controls, the Student’s *t*-test and the Mann–Whitney U test were used. In all the tests, a *p*-value ≤ 0.05 was considered significant. To our knowledge, this was one of the first studies that evaluated FFI-RS in RA patients with inflammatory lesions based on ultrasound. Therefore, we performed a post-hoc analysis to determine the effect size and power of the test. The effect size was d = 0.62 (mean value) and the power of the test was 0.76.

## 3. Results

### 3.1. Patients

We did not observe statistically significant differences in the baseline characteristics between the study and control groups. The baseline characteristics of the patients are shown in Table 1.

### 3.2. Comparison of the FFI-RS Subscale Results between Both Groups

Significant differences were observed between the study and control groups in the FFI total, FFI pain, and FFI stiffness results. The comparison of the FFI-RS subscale results between both groups is shown in Table 2.

### 3.3. Correlation of FFI-RS with HAQ-D

Table 3 shows specifically how the subcategories of commonly used tools (FFI or HAQ) can correlate with each other. The FFI-RS total results correlated with the HAQ-DI total, standing, and walking results. There was also a correlation between the FFI-RS pain results and the HAQ-DI total results in the control group. The FFI-RS stiffness and difficulty results correlated with the HAQ-DI total, standing, and walking results in the study group. Similar correlations were observed in the control group except for the HAQ-DI standing results. The FFI-RS activity limitation results correlated only with the HAQ-DI standing results in the control group. The FFI-RS social issues results correlated with the HAQ-DI total results in both groups, and with the HAQ-DI standing and walking results only in the study group.

The Pearson’s and Spearman’s correlation tests were used, respectively, to parametric or nonparametric assumption.

### 3.4. Correlation between FFI-RS with VAS, DAS 28 and Disease Duration

All the FFI-RS subscale results correlated with VAS in the study group, except for the FFI-RS activity limitation results. In the same group, the FFI-RS total results correlated with the DAS 28 results. There was also a significant correlation between the FFI-RS difficulty results and the disease’s duration in both groups.

The Pearson’s and Spearman’s correlation tests were used, respectively, to parametric or nonparametric assumption.

### 3.5. Comparison between FFI-RS Results with Surface Skin Temperatures of Right and Left Foot Measured by a Thermal Imaging Camera

We performed an analysis of correlations between the FFI-RS total results and the surface skin temperatures of the right and left foot in both groups, and we found no significant correlations.

## 4. Discussion

The incidence of foot problems is strongly associated with the presence of RA [24]. It is approximately two times greater than in the general population [25]. At diagnosis, up to 50% of RA patients already have foot problems [26], and in every third case, foot pain is the reason for presentation [27].

There are not many studies investigating the relationship between ultrasound findings and foot function status in RA patients. For this analysis, we used the Foot Function Index-Revised Short Form Polish Version (FFI-RS-PL), for which relevance in this type of research was previously proven [14,15,20].

Our results show that RA patients, both with and without lesions in the ultrasound, show feet dysfunction.

Michelson (1994) and Otter (2010) suggested that over 90% of RA patients are reported to experience foot problems related to the ongoing disease [24,27,28].

Comparing the patients with lesions in ultrasound examination and those without (Table 2), we noticed that the functional status, as assessed by the FFI-RS total, pain, and stiffness subscales was significantly worse in the control group despite absence of lesions in the ultrasound.

This shows that while the ultrasound’s primary purpose is an objective detection of inflammation to facilitate early diagnosis and follow-up of disease activity [22], it cannot be used to evaluate the functional status of a patient’s foot.

In patients with RA, one of the most popular questionnaires to analyze functional status is the HAQ-DI. However, it does not evaluate the functional status of a patient’s foot, since functional status is subjective and should be measured by a patient-reported outcome measure (PROM), such as the FFI-RS or HAQ. Such a disparity in clinical findings is worth exploring since it will be very important for RA management and evaluation of prognosis. In this study, we demonstrated that the results of the HAQ-DI total, standing, and walking subscales were correlated with the results of some of the FFI-RS subscales. The HAQ-DI standing difficulty results are correlated with the activity limitation and difficulty FFI-RS subscale results, and they are not correlated with the pain FFI-RS subscale results. Such information could guide clinicians to choose the appropriate treatment option (Table 3 and Table 4). This could be used in planning targeted interventions in patient care issues.

Similar results were observed in the study by Ajda Bal et al. 2006. They suggested a need for a more specific evaluation, demonstrating the effects of foot deformities and pathology on the FFI results [29]. Our study showed that the results of the FFI-RS total and all its subscales, correlate (Table 4) with the duration of the disease in both the investigated groups.

On the other hand, the status of a patient’s foot is prone to deterioration as RA progresses, which is related to the disease’s duration. Hence, over time, the foot issues caused by RA become more important than the pain or stiffness in small joints. Some authors [1,5,8,11,27] imply that many structural and biomechanical problems with walking and standing can be caused by the mechanical burden applied to a patient’s weakened musculoskeletal system. This can lead to pain, deformity, shortening of walking distance, reduced activity levels, and worsening of the general well-being.

Our research showed significant correlations between the VAS pain subscale results and the results of several of the FFI-RS subscales like the FFI-RS total, pain, stiffness, difficulty, and social issues (Table 4), especially in the patients with lesions in foot ultrasound. These correlations probably show the intertwining associations between the foot pathologies, as assessed by the FFI-RS subscales, with pain as the main symptom.

Usually, the symptoms are first noticed in the forefoot [10]. Changes in the hindfoot and forefoot anatomy result in an altered foot and ankle motion, higher forefoot plantar pressure, and increased pain on weight bearing and walking.

Moreover, we observed significant correlations between the FFI-RS total and DAS 28 results in the group of patients with ultrasound lesions. There was not any such correlation in the control group (Table 4).

This result is confirmed by the inflammatory lesions observed in ultrasound. This is an interesting observation because few studies analyzed the associations between the foot lesions as observed in ultrasound, the DAS 28 results, and different commonly performed procedures.

Sant Ana Petterle et al. 2013 evaluated 50 patients with RA and 50 healthy individuals using ultrasound and the DAS 28, HAQ, and FFI procedures. As a result, no associations were found between the foot joints’ ultrasound and the DAS-28, HAQ, and FFI results [30].

Baan et al. 2011 evaluated 30 patients with RA using ultrasound, the FFI, and HAQ. As a result, the authors observed a weak but statistically significant correlation between the ultrasound (laser Doppler) of the hindfoot and the HAQ walking results. The authors also suggested that foot function was only weakly associated with the injuries as demonstrated by radiological techniques, and that this association seemed stronger in the hindfoot [31].

Interestingly, the FFI-RS results can be used to discriminate RA patients with a foot pathology among those without ultrasound evidence of foot lesions (Table 2). It is noteworthy that the current practice in assessment of patients with RA involves mainly standard clinical skills of examining signs and symptoms to detect synovial inflammation. Biochemical analyses of the acute-phase response generally reinforce this information, and this data may be combined in a composite score like the Disease Activity Score (DAS28). Early accurate detection and quantification of synovial membrane inflammation are now recognized to be crucial, not only in making a correct diagnosis, but also for the subsequent assessment, management, and prognosis evaluation. The use of ultrasound has a strong justification at this stage [22].

Inflammation is directly associated with the primary long-term outcomes, such as joint damage, functional impairment, and disability. The current diagnostic standards, according to ACR/EULAR, or common tools such as the HAQ or DAS 28, do not analyze the foot problems that occur in over 90% of patients with RA, which may sometimes be the first symptom of the disease. The FFI-RS in its current form is one of the most comprehensive instruments available to assess the functional status of the foot [14], and as our research shows, it correlates with clinical parameters, including the psychosocial aspects of daily living.

The fact that some studies have documented a progression of the joint damage despite an apparent clinical improvement raises important questions [22,32]. As shown by our research, the effects of inflammation, its extent, and duration on the functional assessment conducted to make a diagnosis and monitor the treatment, are not entirely clear. The FFI-RS has an essential role in shifting the paradigm from the reliance on physical and biochemical findings to the use of a patient-relevant diagnostic work-up [14].

We also tried to find a relationship between the FFI total and the foot (ROI) surface skin temperature distribution, but we did not find any significant correlations. Moreover, we found no differences in the foot surface skin temperature between the patients with and without lesions in the ultrasounds. Existence of such differences seemed likely since the thermovision is a useful tool for detecting this type of change [33], but our research showed no such differences.

Our research confirms the effectiveness of using the FFI-RS, but its limitation is a small number of subjects.

## 5. Conclusions

The FFI-RS is an effective tool to assess RA patients’ feet functional status, which affects the general condition of RA patients. In daily clinical practice, specialists should pay attention to the foot problems. The lack of RA-related lesions in the ultrasound of foot joints does not necessarily imply absence of changes in the foot function; therefore, the foot function questionnaires like the FFI-RS should be a tool included in the standard clinical assessment. Our observations require further research in relation with the other imaging techniques for patients with RA

## 6. Limitations to This Manuscript

This is a prospective observational study. Its results are limited to the associations/correlations between the variables of interest. The sample size is small, and the sample selection is based on a convenient sample. Most of the subjects are females, and therefore the findings may not be generalized to males. The participants had a history of 14 years of RA on average and some of them might have been in the remission phase, hence their ultrasonography results were normal (control group). It might be possible that the DAS28 score of 4.17 is representative of painful, burned-out joints, and it might not be caused by having active RA. Therefore, this finding needs further investigation.

## 7. Strength of the Study

This prospective observational study was conducted with a real-time clinical data collection.

## Figures and Tables

**Table 1 jcm-11-02298-t001:** Baseline characteristics of the patients.

	Study Group (with Ultrasound Changes) (*n* = 37)	Control Group (without Ultrasound Changes) (*n* = 24)	*p* ^#^
Age (years)	54.3 (9.5)	57.3 (11.5)	0.27
Gender F/M	32/5	21/3	0.54
Disease’s duration (years)	13.95 (10.4)	13.67 (8.82)	0.96
ESR	12.00 (7.40)	13.33 (9.08)	0.55
Disease status of RA			
DAS 28	4.13 (0.81)	4.17 (0.94)	0.87
Remission			
Mild	0	2	
Moderate	7	1	
Active diseases	26	18	
Comorbidities	4	3	
Hypertension	7	8	
Diabetes mellitus	0	1	
HAQ-DI	2.24 (1.39)	2.68 (1.32)	0.22
VAS (mm)	60.57 (15.86)	62.50 (14.72)	0.63
Surface skin temperature (°C)			
ROI–right foot	30.82 (1.79)	30.55 (2.19)	0.62
ROI–left foot	30.81 (2.05)	30.51 (2.27)	0.60

The results are expressed as mean ± SD; *p* ≤ 0.05. RA- rheumatoid arthritis; NS—non-significant; F—female; M—male; ESR—erythrocyte sedimentation rate; DAS 28—disease activity score 28; HAQ-DI—health assessment questionnaire-disability index; VAS—visual analog pain scale; ROI—the region of interest. ^#^ Either the *t*-test or the Mann–Whitney U test was used to compare the differences between the groups, respectively.

**Table 2 jcm-11-02298-t002:** Differences in Foot Function Index subscale results between the study and control groups.

	Study Group (with Ultrasound Changes) (*n* = 37)	Control Group (without Ultrasound Changes) (*n* = 24)	*p* ^#^
FFI-RS total	61.8 ± 12.9	71.1 ± 17.2	0.020 **
FFI-RS pain	56.1 ± 14.9	66.8 ± 18.2	0.015 **
FFI-RS stiffness	56.9 ± 14.5	65.3 ± 18.0	0.050 *
FFI-RS difficulty	73.3 (± 15.1)	81.2 (± 19.5)	0.079
FFI-RS activity limitation	53.8 (± 20.1)	62.8 (± 25.5)	0.129
FFI-RS social issues	58.1(± 19.7)	68.9 (± 23.2)	0.053

The results are expressed as mean ± SD. ^#^ The *t*-test was used to compare the differences between the groups. * *p* ≤ 0.05 (two-sided); ** *p* ≤ 0.01 (two-sided); FFI-RS—Foot Function Index-Revised Short Form.

**Table 3 jcm-11-02298-t003:** Correlation (r) of FFI-RS subscales with HAQ-DI standing, walking, and total results in the study and control groups.

	Study Group (*n* = 37)	Control Group (*n* = 24)	Study Group (*n* = 37)		Control Group (*n* = 24)	
	HAQ-DI Total		HAQ-DI Standing	HAQ-DI Walking	HAQ-DI Standing	HAQ-DI Walking
FFI-RS total	0.65 **	0.77 **	0.54 **	0.57 **	0.65 **	0.45 *
FFI-RS pain	0.30	0.67 **	0.14	0.29	−0.15	0.30
FFI-RS stiffness	0.49 **	0.63 **	0.37 *	0.51 **	0.08	0.52 **
FFI-RS difficulty	0.70 **	0.75 **	0.60 **	0.60 **	0.23	0.47 *
FFI-RS activity limitation	0.26	0.6	0.38 *	0.24	−0.05	0.30
FFI-RS social issues	0.70 **	0.74 **	0.57 **	0.51 **	−0.04	0.31

* Significant correlation with *p* ≤ 0.05 (two-sided), ** Significant correlation with *p* ≤ 0.01 (two-sided); FFI-RS—Foot Function Index- revised short form; HAQ-DI—health assessment questionnaire-disability index.

**Table 4 jcm-11-02298-t004:** Correlation (r) of FFI-RS-PL subscale with VAS, DAS 28 and disease’s duration in the study and control groups.

	Study Group (*n* = 37)	Control Group (*n* = 24)	Study Group (*n* = 37)	Control Group (*n* = 24)	Study Group (*n* = 37)	Control Group (*n* = 24)
	VAS		DAS 28		Disease Duration	
FFI-RS total	0.48 **	−0.7	0.37 *	−0.16	0.28	0.26
FFI-RS pain	0.43 **	−0.01	0.24	0.03	0.19	0.34
FFI-RS stiffness	0.44 **	0.22	0.30	0.22	0.25	0.12
FFI-RS difficulty	0.37 *	−0.13	0.32	−0.21	0.33 *	0.45 *
FFI-RS activity limitation	0.30	−0.25	0.12	−0.23	−0.04	−0.12
FFI-RS social issues	0.36 *	−0.21	0.32	−0.16	0.14	0.36

* Significant correlation with *p* ≤ 0.05 (two-sided), ** Significant correlation with *p* ≤ 0.01 (two-sided); FFI-RS—Foot Function Index-Revised Short Form; VAS—visual analog pain scale; DAS 28—disease activity score 28.

## Data Availability

The datasets used and/or analyzed during the current study are available from the corresponding author upon reasonable request.

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
