# Peer review of "The Importance of Foot Function Assessment Using the Foot Function Index-Revised Short Form (FFI-RS) Questionnaire in the Comprehensive Treatment of Patients with Rheumatoid Arthritis"

_jcm, 2022, doi:10.3390/jcm11092298_

Round 1

Reviewer 1 Report

  1. Few lines need grammatical and language corrections
  2. Disease status of RA mus be mentioned along with its other variants and systemic illness.
  3. Mention the statistical test for intergroup difference
  4. mean or mean difference of these questionnaires tools must be mentioned in tabular formate in USG positive or negative for feet lesions.
  5. The objective needs to be clear, here it is vague or unclear
  6. How is the sample size estimated, is this sample size is adequate to validate the FFI questionnaire tool
  7. Reliability of each tool also be mentioned 
  8. Details of my comments for correction in attached manuscript.

Author Response

Comments of Reviewer 1

This is regarding the article “The importance of foot function assessment using The Foot Function Index-Revise Short Form (FFI-RS) questionnaire in the comprehensive treatment of patients with rheumatoid arthritis” written by RadosÅ‚aw Rutkowski, MaÅ‚gorzata GiziÅ„ska, MaÅ‚gorzata GaÅ‚czyÅ„ska-Rusin, Magdalena Kasprzak, Elly Budiman-Mak.

Thank you for your comments which are very important. All instructions are taken into account and corrected in the text. We have marked our changes to the manuscript in red.

We place our comments below:

1.Few lines need grammatical and language corrections

We read the text carefully and corrected errors, also the manuscript was checked by an English language specialist

2.Disease status of RA must be mentioned along with its other variants and systemic illness.

In Table 1.  we added information on disease activity according to DAS28

3.Mention the statistical test for intergroup difference

We added information about statistical tests in all tables and changed the NS - (non statistical) abbreviation into test values.

4.mean or mean difference of these questionnaires tools must be mentioned in tabular formate in USG positive or negative for feet lesions.

In Tables 1 and 2 we wrote the sentence that study group is with ultrasound changes and control group is without ultrasound changes

5.The objective needs to be clear, here it is vague or unclear

According to your suggestion, the objective and conclusion have been verified and rewritten more clearly:

In section Introduction we added: Line 63 o 73 “. Our observations show that the changes visible in ultrasound do not reflect the functional state of the feet, which affects the general condition of patients . Therefore, to accurately evaluate the functional changes in the feet, which are known to affect the quality of life in patients with RA, we should use scientifically recognized tools to make the diagnosis process complete. One of these is the Foot Function Index Revised Short Form questionnaire (FFI-RS),
a short version containing 34 questions. This version includes five subscales for pain, stiffness, difficulty, activity limitation and social aspects. According to the authors, the short version has sound psychometric measures [13, 14. 15]. This study aims to prove usefulness  of the FFI -RS Polish Version questionnaire with all of it subcategories  in assessing RA patients' functional status of feet with various clinical changes in feet ultrasound imagine and their correlation with the general condition of the patients.”

In section Conclusion we added: Line 324-330 “The FFI-RS is an effective tool to assess RA patients' feet functional status which affects the general condition of RA patients. In daily clinical practice, specialists should pay    attention to the foot problems. The lack of RA-related lesions in the ultrasound of foot joints does not necessarily imply absence of changes in the foot function; there-fore, the Foot Function questionnaires like the FFI-RS should be a tool included in the standard clinical assessment.  Our observations require further research in relation with the other imaging techniques in patients with RA”

6.How is the sample size estimated, is this sample size is adequate to validate the FFI questionnaire tool.

As to our knowledge, this was the first study to evaluate FFI-RS in RA patients with inflammatory lesions based on ultrasound, so that is why we performed a post-hoc analysis of the effect size and power of the test used. The effect size was d = 0.62 (mean value) and the power of the test was 0.76. We found these values sufficient for the size of our research group.

We added this information in part of Statistical analysis:

Line 160 to 164 “As to our knowledge, this was the one of the first study to evaluate FFI-RS in RA pa-tients with inflammatory lesions based on ultrasound, so that is why we performed a post-hoc analysis of the effect size and power of the test used. The effect size was d = 0.62 (mean value) and the power of the test was 0.76”

7.Reliability of each tool also be mentioned

We thought that the tools used in our research are commonly used by many authors, therefore we described them poorly. After your suggestions, we added information about their reliability in section  Data collection:

Disease Activity Score 28. Line 93 to 94  ……”Response criteria have been extensively validated and are finding wide range of applications in RA clinical trials and to monitor individual RA patients”…… Line

Health Assessment Questionnaire-Disability Index. Line 100 to 105  ….”The HAQ-DI is a predictive factor of future disability and joint damage in patients with RA. Because it demonstrated sensitivity to change, the HAQ-DI was recommended by the American College of Rheumatology (ACR) to be incorporated into the core set of outcome measures of RA disease activity . The HAQ-DI not only is considered an essential measure of disability in patients with RA in clinical trials, but also is used in clinical practice.”…. Line

The Visual Analogue Pain Scale. Line 108 to 111 “The visual analog scale (VAS) is a valid and reliable measure of chronic pain intensity. This is  a simple and commonly used tool to use by anyone cognitively capable of understanding the parameters and responding to clinician’s instructions”….. Line

Foot Function Index Revised –Short Form Polish Version. Line 119 to 123  ….. “Each participant was coached to correctly fill the questionnaire. Scores range from 0-100, with higher scores representing a worse foot function.  This questionnaire is a reliable and frequently used tool in patients with osteoarthritis and rheumatological diseases. In 2017 we adaptation and validation the Foot Function Index-Revised Short Form into Polish. The person reliability of Polish version was 0.95. Line

Ultrasound. Line 124 to 126 “Ultrasound is a quick and inexpensive way to detect synovitis, tenosynovitis, tendon tears and bursitis. However, it is operator-dependent and also depends on the quality of a USG”…. Line

Thermovision. Line 133 to 136  “Thermovision techniques have been used for various purposes in medicine, including clinical testing of drugs , assessment of vascular reactions in hands, diagnosis of Raynaud’s syndrome, evaluation of observational changes in osteoarthritis, detection of diferent kinds of tumors ,assessment of the skin condition in diabetic feet”….

In response to your question:

“What is the purpose of this table.  correlation matrix of these parameters not related to your research question. are you evaluated these questionnaires?

No where reliability and sample adequacy are mentioned adequacy

In Table 3. we want to show how specifically the subcategories of commonly used tools (FFI or HAQ) can correlate with each other  and that the changes in ultrasound or their absence do not precisely indicate the functional state of the feet.

In 2017 we adapted and validated the Foot Function Index-Revised Short Form into Polish.

“Biomed Res Int. 2017;2017:6051698. doi: 10.1155/2017/6051698. Epub 2017 Nov 27.”

Section References:

  1. Rutkowski, R., Gałczyńska-Rusin, M., Gizińska, M., Straburzyński-Lupa, M., Zdanowska, A., Romanowski, M. W., Romanowski, W., Elly Budiman-Mak & Straburzyńska-Lupa, A. (2017). Adaptation and validation of the foot function index-revised short form into Polish Biomed Res Int 2017;2017:6051698.

Authors are requested to measure co-variance AS MANY OF OF THE ITEM OF QUESTIONNAIRE ARE RELATED.

The main goal of our study was to demonstrate the usefulness of foot function tests in the overall assessment of RA patients, so the statistical analysis used in the study focused mainly on demonstrating the effectiveness of using the questionnaires to assess the functional status of the foot, regardless of the ultrasound image. Since the forms used for the study are validated and complete tools, the analysis of co-variance could yield uninterpretable data, disrupting the chosen purpose of the study.

With regards,

Authors

Reviewer 2 Report

Thank you for your manuscript, I am glad to see the changes made to the previous manuscript.

The study carried out in patients diagnosed with RA provides evidence on the way to a better diagnosis and a better forecast of the disease.

It is a study with a high degree of limitations that must be resolved in future studies, having to include a larger sample and carried out over a longer period of time. If you are going to carry out a validation in a population with this specific disease, you must carry out some statistical test that is complementary to the one already carried out in order to analyze the true reliability and validity in patients with this pathology.

Having adapted the FFI-SF questionnaire to the Polish population does not mean that it is valid for the different diseases.

The change you would make would be in Lines 114 to 118: do not repeat twice the measurement and assessment procedure of the Visual Analogue Scale from 0 to 100, explain what was done in your study.

Thanks again for your manuscript. 

Author Response

Dear Reviewer, we are very grateful for your careful study of our manuscript. Thank you for your valuable comments and helpful suggestions. We hope that the corrections we have made in manuscript “The importance of foot function assessment using The Foot Function Index-Revise Short Form (FFI-RS) questionnaire in the comprehensive treatment of patients with rheumatoid arthritis” written by RadosÅ‚aw Rutkowski, MaÅ‚gorzata GiziÅ„ska, MaÅ‚gorzata GaÅ‚czyÅ„ska-Rusin, Magdalena Kasprzak, Elly Budiman-Mak.are appropriate and satisfactory for you.

Our changes we include  below:

1.We have removed the duplicate text in the lines from 118 to 120

The VAS pain scale results were obtained by measuring the distance in millimeters from the beginning of the scale to the position selected by the patient from “0” to “100 mm”, where “0” is for "no pain," and “100” is for "the worst possible pain

  1. At the section Materials & Methods, Participants, line from 79 to 83 we added following text “In the basic characteristics for each of them, we have defined, among others, the following data: Age, Gender, Disease's duration, Comorbidities DAS 28, HAQ-DI and VAS. Additionally, we performed Feet Surface skin temperature analyzes. The next step of research procedure was examination with used the FFI-RS questionnaire. The obtained results were statistically analyzed
  2. In line 192 “Comparison/Correlation of FFI-RS with HAQ-D” we removed the word “Comparison”

With regards,

Authors

Reviewer 3 Report

Authors should pay attention to the following aspects:

In the Abstract: 

Line 36: the VAS abbreviation should be clarified before used.

In the Results:

Lines 139-142: What is the baseline characteristics? Only the characteristics of the patients in table 1 are baseline characteristics? So the results of table 2 are what?

The results should be rewritten more simply and clearly.

In the Discussion:

Line 190:   "is ca. two" - error...

Line 220: "in the study by Ajda Bal et al." - the date is missing!

Line 246: "Sant Ana Petterle et al. " - the date is missing!

Line 282-283: ... "our research shows that they do not affect the foot function..."  Be carefull with this statement!  You didn't do mechanical gait analysis (kinematics and kinetics). Your results did not allow you to make these observations.

You predominantly used qualitative tools, when you should have used simultaneously, for example, plantar pressure insoles and force platforms, among others.

Author Response

Comments of Reviewer 2

This is regarding the article “The importance of foot function assessment using The Foot Function Index-Revise Short Form (FFI-RS) questionnaire in the comprehensive treatment of patients with rheumatoid arthritis” written by RadosÅ‚aw Rutkowski, MaÅ‚gorzata GiziÅ„ska, MaÅ‚gorzata GaÅ‚czyÅ„ska-Rusin, Magdalena Kasprzak, Elly Budiman-Mak.

Thank you for your comments which are very important. All instructions are taken into account and corrected in the text. We have marked our changes to the manuscript in green.

We place our comments below:

In the Abstract:

Line 36: the VAS abbreviation should be clarified before used.

We corrected it

In the Results:

Lines 139-142: What is the baseline characteristics? Only the characteristics of the patients in table 1 are baseline characteristics? So the results of table 2 are what?

Table 2. shows the differences between the groups in the FFI-RS questionnaire and its subcategories. We expanded the results to include difficulty, activity limitation and social issues to show the entire questionnaire. Our main assumption in this table is to show the differences in FFI-RS. We did not want to characterize the groups based on this.

The results should be rewritten more simply and clearly.

We improved the tables and added all the subcategories of FFI-RS, also the name of the statistical test and its value

In the Discussion:

Line 190:   "is ca. two" - error...

We corrected it

Line 220: "in the study by Ajda Bal et al." - the date is missing!

We corrected it

Line 246: "Sant Ana Petterle et al. " - the date is missing!

We corrected it

Line 282-283: ... "our research shows that they do not affect the foot function..."  Be carefull with this statement!  You didn't do mechanical gait analysis (kinematics and kinetics). Your results did not allow you to make these observations.

We analyzed this sentence and changed it according to your suggestions

Line 319 to 320:  “Existence of such differences seemed likely since the thermovision is a useful tool for detecting this type of changes [35] but our research showed no such differences.”

You predominantly used qualitative tools, when you should have used simultaneously, for example, plantar pressure insoles and force platforms, among others.

This is an extremely important suggestion and we will apply it in our future research.

With regards,

Authors

Reviewer 4 Report

The article deals with a potentially interesting topic: foot involvement as a consequence of rheumatoid arthritis.

A statistical comparison between two patients with proven lesion and a control group is proposed. The statistical analysis is adequate and leads to the expected result that the FFI-RS can discriminate between the two groups, which is interesting but not surprising.
I feel that the work while impeccably performed and not questionable in its methods, does not present a broad scientific interest, in the opinion of this reviewer. My impression is that the work, as the authors themselves admit in the conclusions, is limited in scope and preliminary as they acknowledge.

I am asked if I detect unwarranted self-citations, I think the way the reference [35] is used might just be a ruse to include a citation, but it doesn't really seem too justified or connected to the actual work.

Author Response

(The authors gave the same response as above.)

Round 2

Reviewer 1 Report

Satisfied with the present form

Author Response

(The authors gave the same response as above.)
